# Healthcare Quality from the Perspective of Patients in Gulf Cooperation Council Countries: A Systematic Literature Review

**DOI:** 10.3390/healthcare12030315

**Published:** 2024-01-25

**Authors:** Nizar Alsubahi, Milena Pavlova, Ahmed Ali Alzahrani, Ala’eddin Ahmad, Wim Groot

**Affiliations:** 1Department of Health Services Research, Care and Public Health Research Institute—CAPHRI, Maastricht University Medical Center, Faculty of Health, Medicine and Life Sciences, Maastricht University, P.O. Box 616, 6200 MD Maastricht, The Netherlands; m.pavlova@maastrichtuniversity.nl (M.P.); ahmed.alzahrani@maastrichtuniversity.nl (A.A.A.); w.groot@maastrichtuniversity.nl (W.G.); 2Department of Health Service and Hospital Administration, Faculty of Economics and Administration, King Abdul Aziz University, Jeddah 21589, Saudi Arabia; 3Digital Marketing Department, Al-Zaytoonah University of Jordan, Amman 11733, Jordan; a.ahmad@zuj.edu.jo

**Keywords:** patient-centered care, patient perceptive, health care quality, Gulf Cooperation Council countries, systematic review

## Abstract

With the increased focus on patient-centered care, consensus on healthcare outcomes of importance to patients becomes crucial. Based on a systematic review of the literature, this study confirms the perspectives of patients on healthcare quality in GCC countries. Online databases were searched for relevant peer-reviewed articles published from 2012 to 2023. Twenty-two articles retrieved from the search were qualitatively analyzed based on the Preferred Reporting Items for Systematic Reviews and Meta-Analysis (PRISMA) guidelines. Most articles (90%) reported studies conducted in Saudi Arabia. Patients in GCC countries face common problems in the care delivery process, which contribute to negative perceptions of quality. These problems include diagnostic and medication errors, provider–patient communication problems, missed appointments with physicians, problems in emergency care access due to geographical distance and transportation barriers, long waiting times, and physical environments. Notably, healthcare quality is perceived to be an outcome of multiple factors dependent on the location and category of healthcare service providers; for instance, disparities in perceptions of quality were observed between patients attending Primary Health Care (PHC) centers in rural and urban areas. Issues such as lack of equitable healthcare delivery and deficiencies in Emergency Medical Services (EMS) effectiveness were disparately recognized as quality concerns by different patient populations. The findings provide insights into healthcare quality and area of weakness needing strategies and policies to ensure patient-centered, safe, equitable, timely, and effective healthcare. Healthcare providers and policymakers in GCC countries can use the results to plan, assess, and improve care delivery. Trial registration: PROSPERO ID: CRD42022326842.

## 1. Introduction

Many healthcare systems aim to become more value-based and to focus on outcomes relevant to patients and patient-centered care. This objective comes with a shift in emphasis from quantity to quality of care [1]. Quality is a vital and core element of healthcare because the lack of it can lead to profound consequences and harm to patients. Additionally, quality healthcare services can enhance the living and working conditions of individuals, families, and communities, reduce inequality, support human development, improve health and well-being, and foster healthy and resilient communities and economies [2]. According to the World Health Organization (WHO), quality refers to the degree to which health services for individuals and populations increase the likelihood of the desired health outcomes [3]. Furthermore, the Institute of Medicine (IOM) emphasized the critical need to redesign healthcare to meet the demands of the twenty-first century [4]. This includes ensuring a continuous healing relationship that considers individual patients’ needs and values, and focuses on patients’ health and well-being rather than their illness [4,5]. To render quality as the focus of attention, a need emerges for a common understanding and framework of quality.

The literature proposes various frameworks for analyzing quality of care. Donabedian, the IOM, and LoBiondo-Wood et al. present a quality framework with six domains of quality, namely, safety, timeliness, efficiency, effectiveness, equitability, and patient-centeredness [6,7]. Safety is concerned with the extent to which healthcare services prevent harm to patients. Timeliness ensures that patients receive care when needed without lengthy or unnecessary waiting and harmful delays. Efficient care ensures optimal utilization of resources to avoid waste of equipment, energy, supplies, and ideas. Effective care is based on scientific knowledge and evidence of the benefits for patients or consumers, while avoiding misuse and underuse. Equitability ensures consistency in care delivery, such that no variance exists based on personal characteristics, including socioeconomic status, ethnicity, gender, religion, and geographic location [6,7]. The abovementioned framework demonstrates that quality is a broad and complex concept.

Moreover, the perception of quality is subject to aspects that patients consider most significant in healthcare services and the healthcare system. In effect, the perception of care among patients varies across patients [8]. For certain patients, timeliness may not be important, although safety may rank higher for others. Patient-centered care respects the unique values of patients and is responsive to individual needs and preferences [6,7]. Measuring and evaluating patient-centered healthcare can help improve patient outcomes and enable data collection for national and international comparison. This can facilitate comparative studies and a greater understanding of the correlates, predictors, and outcomes of person-centered healthcare services.

In the Gulf Cooperation Council (GCC) countries (namely, Saudi Arabia, the United Arab Emirates (UAE), Kuwait, Qatar, Bahrain, and Oman), quality of care and value for patients are slowly penetrating the healthcare system. However, healthcare systems in the GCC are facing various challenges. For example, the region continues to record a high number of medication errors, which may suggest low-quality safety [9]. Compared to other countries in Europe and North America, for example, quality has been of less importance in the GCC region, and the quality outcomes have been more negative [9]. The areas of focus for the governments of the GCC countries would, thus, be different from those of other countries, concerning care quality. For instance, a study in Brazil showed that it is important to address the population’s health needs and problems, indicating that in this country, the priority areas are not as obvious as in the GCC [9]. Thus, improving healthcare quality is a national priority for GCC governments in fulfilling the present and future health needs of the population [10]. This role can, however, only be accomplished by focusing on the quality areas in research, since quality has emerged as one of the most important factors influencing patients’ choices between government and private hospitals in the GCC countries. Service quality is essential for health organizations to remain competitive and sustainable [11]. As such, research into healthcare quality is needed to enhance quality care in the GCC and place improvements in care quality in the GCC region on a par with those in other countries.

Previous research in GCC countries indicates that the thoughts and satisfaction of patients play a crucial role in their assessment of the quality of the service they receive [12]. The GCC countries have a mixed healthcare system that includes both public and private providers, with the government playing the primary role in funding and regulation [13]. Furthermore, they have private health insurance providers, which primarily serve foreign citizens, who constitute an average of 70% of the total population in these countries [14]. Through their respective public health systems, GCC countries manage a universal healthcare system that provides their citizens with free or subsidized medical services [13,14]. On the other hand, foreign nationals residing or working in GCC countries are typically required to obtain private health insurance or an insurance through their employers. Although the healthcare systems in the GCC countries are similar, there is a subtle difference in the level of government investment in the healthcare sector. For example, Bahrain, the UAE, and Saudi Arabia make significant investments in their healthcare sectors to improve care quality and position themselves as medical tourism destinations [15,16]. Studies are necessary to evaluate the impact of these efforts and healthcare transformations focusing on attracting medical tourists.

This study aims to review evidence on the six domains of healthcare quality from the perspective of patients in GCC countries. The domains include safety, timeliness, efficiency, effectiveness, equitability, and patient-centeredness [6,7]. Reviewing evidence on these six domains provides insights into the performance of the healthcare systems in the region. These factors are interdependent, which makes it worth examining their coupling effects as part of research into healthcare quality. For instance, timeliness can be a critical factor in patient safety, as are both effectiveness and efficiency in service delivery. Consequently, the findings will inform efforts toward quality improvement exerted by healthcare providers and policymakers. The six domains are useful for examining healthcare quality via analysis of the structure, process, and outcomes of health services [6,17]. To the best of our knowledge, this is the first systematic review conducted in the last 11 years that examines healthcare quality in the GCC countries from the patients’ perspective.

## 2. Methodology

This systematic review was guided by the Preferred Reporting Items for Systematic Reviews and Meta-Analysis (PRISMA) 2020 checklist [18]. PRISMA is recognized as an effective tool for evaluating systematic reviews and meta-analyses by researchers in different fields, as well as by journal peer reviewers and editors. The review is based on a set of evaluation criteria highlighted in a checklist that is evidence-based [18]. The checklist includes additional reporting guidance for each of the evaluation items, along with reporting exemplars. The recommendations and guidelines are widely endorsed and have been adopted in more than 60,000 reports and approved by more than 200 journals and organizations for systematic reviews [18]. The PRISMA framework has also been adopted for application in multiple disciplines ranging from social sciences to arts [18]. As such, it is a suitable framework for a systematic review. The Appendix A provides the checklist for this review. The collection and review of articles were conducted between 2012 and 2023, and a final search was conducted in November 2023. All authors developed and agreed on the review protocol, which was registered in the international prospective register of systematic reviews (PROSPERO) on 30 May 2022, under the following ID: CRD42022326842 (CRD: Centre for Reviews and Dissemination).

The review process was conducted over four steps, namely, study selection, assessment of quality studies, data extraction, and data synthesis and reporting. These steps contribute to the manuscript and research in general. Study selection determines the quality of data that can be collected and, subsequently, the research credibility, while assessment of the quality of studies ensures that the articles ultimately selected for inclusion are reliable and credible sources of information. Data extraction leads to the actual data and information for reporting, while synthesis and reporting contribute to the research, as this step entails the presentation of the study findings. These steps are further described in subsequent sections.

### 2.1. Sources and Search Terms

A search was conducted in PubMed, Scopus, Cumulated Index to Nursing and Allied Health Literature, Business Source Complete, APA PsycINFO (EBSCO), and Social Index. Table 1 presents the key terms used in the search, based on the review objective. To identify relevant articles, Medical Subject Headings were used, and their combinations were evaluated using Boolean operators (AND, OR) to optimize the search strategy. Synonyms and differences in spelling were also considered. We manually searched the reference lists of relevant articles to identify potential new studies that were missed during the initial search. Synonyms and differences in spelling were also considered. The Appendix A documents the exact search phrase used in the database.

### 2.2. Inclusion and Exclusion Criteria

The study employed the patient, intervention, comparison, outcome, and time (PICOT) framework to determine the inclusion and exclusion criteria that can be shown in Table 2.

### 2.3. Study Selection

The study used the bibliographic reference management software EndNote X20 to store titles and abstracts from the initial search. Subsequently, duplicate entries were omitted. Two authors (N.A. and A.A.) independently screened for potentially eligible articles based on the title and abstract, which comprised the first step in the process. The second step involved a screening based on the full text, using the inclusion and exclusion criteria to refine the search. The two authors independently screened the full text of potentially eligible articles for relevance. In the final step, they screened the reference lists of publications selected based on the full text.

### 2.4. Assessment of Quality of Studies

Quality appraisal is essential in systematic reviews as it evaluates the methodology and potential biases in study design, conduct, and analysis. The critical appraisal tools of the Joanna Briggs Institute (JBI), developed and approved by the JBI Scientific Committee, are widely used for this purpose [19,20]. In this review, the quality of the included studies was assessed using JBI critical appraisal tools specifically designed for analytical cross-sectional studies, employing a checklist format. We used the Critical Appraisal Skills Program (CASP) checklist for qualitative study [21], and the Mixed Method Appraisal (MMAT) Tool was used to mix method studies [22]. Studies were rated based on these tools, and the overall quality rating percentage was calculated. The studies were then categorized as “good” (score: 66–100%), “fair” (34–65%), and “poor” (0–33%). Table A1, Table A2 and Table A3 show a quality assessment for the included studies, and can be found in Appendix B.

### 2.5. Data Extraction

Data extraction was performed by two independent reviewers (N.A. and A.A.), using a standardized data collection Excel sheet. The extracted data, which included information on study year, author, aim, quality dimension, sample size, country, and main findings, were subsequently verified by a third reviewer (M.P. and W.G.). Table A4 in Appendix B provides the literature review matrix for the included studies.

### 2.6. Data Synthesis and Reporting

We summarized the extracted data using a narrative synthesis approach to provide an overview of the perceptions of patients in GCC countries with regard to healthcare quality. The findings were synthesized and reported with respect to six healthcare quality domains: safety, timeliness, efficiency, effectiveness, equitability, and patient-centeredness [6,7]. It should be noted that the research followed the Preferred Reporting Items for Systematic Reviews and Meta-Analyses (PRISMA) guidelines in the reporting of this systematic review. A meta-analysis was not performed, due to the wide heterogeneity of the included studies.

## 3. Results

### 3.1. Search Results

The initial search identified a total of 1201 publications, out of which 115 were considered potentially eligible by screening the titles and abstracts and were retrieved for full-text review. Afterward, twenty-one studies met the inclusion criteria after full-text screening, and one study was further included after reference screening. Thus, 22 studies were included and underwent critical appraisal and analysis (Figure 1).

### 3.2. General Description of the Selected Articles

Table 3 presents the main characteristics of the selected articles, most of which were published from 2012 to 2013 (n = 5; 23%). Table 3 also provides a summary of the data collection methods of the 22 articles and indicates that quantitative design was the most frequently applied method (n = 19; 86%), while the two remaining were qualitative (n = 2; 9%), and the last one was qualitative (n = 1; 4.5%). It also outlines the characteristics of the key variables of interest. Nineteen articles reported on patient-centered care (86%), six on timelessness (27%), two on safety (9%), one on effectiveness (4.5%), and one on equity (4.5%). The majority of publications used data from Saudi Arabia (90%), and 10% were from UAE. No studies were conducted in Oman, Qatar, or Bahrain. The sample sizes ranged from a few dozen to a few thousand participants. Only one study included a sample size of <100 respondents. Most studies included 100 to 500 respondents (n = 12), followed by 501 to 1000 (n = 8) and >1000 (n = 1).

### 3.3. Summary of Findings Related to Healthcare Quality Domains

#### 3.3.1. Safety

Two studies on safety were conducted in Saudi Arabia; in a study by Alasqah, various safety problems were identified, with vaccine-related issues accounting for 11% and diagnosis-related issues accounting for 27% of all problems. Among the perceived safety problems, diagnostic error was the most commonly reported, accounting for 26.7% of the cases. Communication issues followed closely at 24.1%, whereas medication errors accounted for 16.3%. The reported consequences of harm included financial problems (40%), increased care needs (32.4%), physical health issues (32%), limitations in activities (30.6%), increased healthcare needs (30.2%), and mental health concerns (26.8%) [23]. The highest average scores for patient-centered care were attained by two items, namely, “A place where I feel safe” and “A place that is neat and clean”, which were related to safety [24].

#### 3.3.2. Timeliness

Six studies reported on waiting time. Ref. [25] introduced issues that led to the highest waiting time and linked these problems to software compared within the outpatient management software. These issues included appointment-type problems in scheduling software, ticket numbering problems in queuing software, physician clinic reporting in time attendance software, early arrival of patients in queuing software, and missing flow problems (physician distribution list) in scheduling software.

One study based in the UAE by [26] demonstrated a significant difference in patient-reported reasons for using private transportation between two cities in the UAE, namely, Abu Dhabi and Al Ain (*p* < 0.01). The residents of Abu Dhabi and Al Ain were more likely to perceive that private transportation is faster (50.9%) and easier (27.8%), respectively. The authors observed no significant difference in reasons between those using private transportation and those who were transported by ambulance (*p* = 0.75).

According to [27], a substantial 68% of respondents from government hospitals emphasized the importance of time as a critical aspect of the service that they received. The findings indicate that timely service and appointments are key factors contributing to patient satisfaction at these hospitals [25].

Moreover, Ref. [28] (Saudi Arabia) found that half of the patients reported satisfaction with the waiting time, whereas the other half were dissatisfied with the intervals between arrival and registration and between registration and consultation, consultation time, and overall waiting time. More than 90% of the dissatisfied patients waited >20 min between arrival and registration, whereas only 2% of family medicine patients waited >20 min between arrival and registration. The study pointed to the lack of a significant association between patient waiting times and satisfaction scores. Ref. [29] also conducted another study in Saudi Arabia, and found that the majority projected the estimated time of arrival of an ambulance to their homes to be approximately 30 min or more, and 94% reported the need for MEDEVAC (air ambulance). According to a study conducted by [30], a significant number of patients expressed dissatisfaction with the waiting time. Specifically, 38.5% of the participants reported experiencing long waiting periods.

#### 3.3.3. Efficiency

None of the studies included any relevant information on this domain.

#### 3.3.4. Effectiveness

One study reported that patients perceived Emergency Medical Services (EMS) effectiveness as having a deficit to address. Over one-third did not know the number to call in case of emergency, and 17.7% would prefer not to have the presence of a male paramedic in the case of a medical emergency involving women. All of this could affect the effectiveness of EMS [26].

#### 3.3.5. Equity

One study reported on equity. Ref. [27] revealed notable disparities among the various aspects of the respondents attending Primary Health Care (PHC) centers in both urban and rural areas. The study also highlighted the lack of equitable healthcare delivery, as shown in the significant disparities between rural and urban PHC patients in receiving treatment with dignity, respect, and understanding of the treatment [27].

#### 3.3.6. Patient-Centered Care

In all, 19 studies reported on patient-centered care. A study comparing Jordanian and Saudi healthcare quality from patients’ perspectives found significant differences in the quality of healthcare services between the two countries. Accessibility was better in Saudi Arabia’s private hospitals [28]. A study reported in [31] highlighted nine vital patient-centered care dimensions for patients. All participants agreed that accessibility, minimized cost, information, education, and staff attitudes and communication were important, and the other five aspects highlighted were staff competence, privacy, physical comfort, psychological and emotional support, and continuity and coordination of care [29]. A study reported in [32] found that 35.6% of patients expressed high satisfaction with healthcare quality. Non-Saudi retired males living in major cities and those who paid for their healthcare services in cash reported the lowest level of satisfaction [30]. Furthermore, satisfaction was found to be strongly related to insurance coverage. Another study [33] found that the highest degree of satisfaction among patients comes from communication (clear discharge instructions and updating the patient’s family with changes). However, [34] found that most patient dissatisfaction comes from poor communication and needing help reading materials provided by the physician, in contrast with [35], who reported that communication resulted in the lowest degree of satisfaction, and that the nursing professional domain produced the highest degree of satisfaction, instead. Mohamed et al. identified the highest degree of satisfaction with primary healthcare services (81.7%), and the cleanliness of the facilities (33.1%) and the technical competence of the staff (24.2%) were the primary factors that influenced this satisfaction. However, unsuitable buildings were the most commonly cited for dissatisfaction [36].

The highest degree of satisfaction was found for access to healthcare. Ref. [30] examined patient satisfaction with care in Makkah, Saudi Arabia, and found that approximately 86% of patients were satisfied with the distance between their home and the nearest clinic; 46.5% agreed that the working hours at the clinic were suitable for them; 37% found difficulty in obtaining an appointment; and 48.5% were dissatisfied with access to healthcare services [31]. In addition, Ref. [33] found that patients across all primary healthcare services (PHCSs) reported high levels of satisfaction. However, notable differences were observed for the comparison of urban and rural respondents, in particular in relation to education level, monthly income, medical investigations, timely provision of blood tests, extended opening hours, distance, cleanliness, and preventive healthcare [27]. This is also supported by [36] Hamam et al., who reported that 50.5% of patients whom they surveyed were satisfied with the rapport that they had with their physicians. Various factors, including age, level of education, the presence of a chronic illness, appointment status, professional status of physicians, and physicians having a nonsurgical specialty, were significantly (*p* < 0.0001) associated with this satisfaction [32]. On the other hand, another study that investigated the care satisfaction of patients with diabetes found higher rates of dissatisfaction in patients aged <50 years, in males, in those with low education levels, and in those with high monthly incomes [37].

Aldossary found that the overall patient satisfaction with dental settings was 3.61 out of 5.0 (72.2%). The highest satisfaction score was received for personal issues with the dental clinic domain (3.93/5; 78.6%), whereas the least satisfaction score was for access to the dental clinic domain (3.29/5; 65.8%). Among all items, the cleanliness of the facility showed the highest score for satisfaction (4.11/5; 82.2%). The least satisfaction was for the ease of contacting the dental clinic (2.71/5; 54.2%) [34].

Al Momani and Al Korashy found that among the individual items for nursing care, information obtained the lowest satisfaction score among patients in Saudi Arabia [38]. A study conducted in Dubai by Mahboub et al. found that patients particularly valued a clear explanation from doctors and nurses; some patients complained about the lack of quality customer care, and encountered difficulties when attempting to communicate with hospitals and clinics over the phone [25]. In addition, two studies found that 55% and 33% of participants in the UAE [26] and Saudi Arabia [36], respectively, stated that they did not know the emergency telephone numbers.

Al Momani and Al Korashy revealed the negative experiences of patients in Saudi Arabia with nursing care in terms of information, caring behavior, and nurse competency and technical care [38]. Alotaibi et al. also conducted a study in Saudi Arabia, and mentioned that the availability of drugs in hospitals was dissatisfactory [39]. Bokary’s study was intended to determine the factors that affect patient satisfaction, identify patients’ unmet healthcare and informational needs, and suggest measures to fill these gaps in healthcare systems. According to the study, patients expressed a high level of satisfaction, rating the medical information subscale as “very good” and the relationship subscale as “excellent”. However, it was also evident that certain aspects of their healthcare required further attention. Therefore, stakeholders should prioritize addressing the unmet healthcare and informational needs of patients as a means of enhancing overall patient satisfaction [40]. A study by Al-Sahli et al. found that in Saudi Arabia, most patients (84.3%) reported that they preferred to be treated at a governmental hospital, and the results suggested a significant association between patient characteristics and their perspectives with respect to person-centered care; the participants in this study perceived their healthcare environments as a highly person-centered climate of care [24].

#### 3.3.7. Systematic Map

Table 4 shows a systematic map of the problems identified and the measures that are being taken to address them, as found in the literature. It should be noted that in most cases, the literature reviewed fails to propose possible solutions to the problems identified as healthcare challenges, classifying them into distinct categories and proposing corresponding improvements. Safety concerns, such as medication and diagnostic errors, are targeted with suggestions to enhance safety and cleanliness. Timeliness issues, including scheduling problems and prolonged waiting times, recommend the use of private transportation. The effectiveness category focuses on strengthening emergency services, while equity concerns advocating for strategies to address healthcare disparities. Patient-centeredness challenges prompt the adoption of strategies to enhance the overall patient-care experience. This comprehensive overview highlights key areas for improvement in the healthcare system, addressing safety, timeliness, effectiveness, equity, and patient-centeredness.

#### 3.3.8. Summary of Conclusions

Table 5 presents a summary of the conclusions drawn from the study with regard to the different factors in healthcare quality. The table shows that a comprehensive overview of healthcare quality domains, revealing challenges in safety, timeliness, effectiveness, equity, and patient-centered care. The findings underscore the importance of addressing these issues to enhance overall healthcare quality and patient satisfaction.

## 4. Discussion

The objective of this review was to cite evidence on healthcare quality in GCC countries from the patient’s perspective, including a lack of patient-centered care, ineffective communication by providers, lack of information, long waiting times, and safety problems and access problems due to geography and transportation barriers. Studies by Alhawary and by Yousef, et al. identified the highest satisfaction with access to healthcare [28,41]. However, Atallah et al. and Aldossary et al. found dissatisfaction in patients with respect to access [34,35]. Specifically, the study identified several factors influencing patient satisfaction with healthcare in GCC countries, including the physical environment (i.e., quality of buildings and availability of parking space), communication, provider behavior (e.g., language), waiting time, care accessibility, and comfort. In their investigations of patient satisfaction in Saudi Arabia, Al-Sahli et al. and Fozan reported positive experiences regarding providers’ respect for religion and culture, their competence, and environmental safety and hygiene [24,33]. However, Atallah et al., Al Momani, Al Korashy, and Al Fozan reported patients’ dissatisfaction with the provider–patient communication and discharge instructions [33,35,38]. Faqeeh et al. and Al Ali and Elzubair found good communication between physicians and patients [27,32]. Moreover, Mahboub et al., found that patients in private clinics receive a clear explanation from doctors [25]. Many studies reported that patients were dissatisfied with the waiting time [31,42,43]. These studies reported negative experiences with waiting time—a common issue. Efforts to improve care quality, patient outcomes, and satisfaction should focus on the shortcomings identified in these areas.

A similar result supported by Hobani on the care quality of gestational diabetes in Saudi Arabia supported the findings of this review. Through a systematic review and qualitative patient interviews, Hobani identified several barriers to high-quality gestational care from the perspective of Saudi women, including limited care access, high cost, communication problems, low provider competence, and long waiting times [44].

Patients in the selected studies identified several concerns related to healthcare quality, across the six dimensions. We found that most articles included emphasized patient-centered care, which aligned with the broader literature, underlining its critical role in healthcare quality [45,46]. A comparative study between Jordan and Saudi Arabia reinforces the idea that cultural and contextual factors influence patient experiences. Accessibility emerged as an essential factor, with Saudi Arabia’s private hospitals outperforming those of Jordan. This finding resonates with those of Webair et al. [29], who identified accessibility and minimized cost as vital for improving patient-centered care.

Previous studies highlighted the role of cultural and contextual factors in influencing care quality and satisfaction [47,48]. The discrepancies in patient satisfaction regarding communication, as highlighted by two included studies [33,35], emphasize this aspect’s complexity. The highest satisfaction was found in one study to come from communication, while in another it was reported as the lowest. These variations may be attributed to differences in communication styles, patient expectations, or healthcare settings. As suggested by Alfaqeeh et al. [27] and Al-Ali et al. [32], the role of cultural and linguistic factors underlines the need for personalized and culturally competent communication strategies. Evidence shows that actively listening to patients, respecting their cultural values, and involving them in decision-making fosters trust in healthcare providers and effective partnerships with the patients, improving health outcomes and reducing health disparities [49]. Another study from the USA found that communication and accessibility were the topmost aspects of care quality, as identified by patients [50]. The same findings were also reported by a previous systematic review of articles evaluating patients’ perspectives toward patient-centered care in the USA, and the remaining were from India, Spain, Sweden, Asia, Pakistan, Turkey, Mauritius, and Central America [51].

Our findings indicate that specific domains, such as cleanliness and access, significantly impact overall satisfaction, which suggests that further extensive and detailed studies are required to explore the multifaceted nature of patient experiences and satisfaction with healthcare quality. Previous studies that were conducted outside the GCC region align with our findings, showing that access to healthcare and cleanliness are major factors in care quality [52,53].

Regarding safety, we found that patients were most concerned about the prevalence of diagnostic errors, communication issues, and medication errors. Diagnostic errors were the most commonly identified safety concern, with potential financial, physical, and mental health consequences for affected patients. Similar findings were also reported in other previous studies [54,55], as was the association between safety and patients’ feelings of being welcome and the cleanliness of healthcare settings, as reported by the included study of Al-Sahli et al. [24], which further underscores the interconnectedness of safety and patients’ overall experiences.

Equity in healthcare delivery implies the importance of fair and dignified treatment for all patients. This systematic review showed disparities in various aspects, including education level, monthly income, and timely provision of services. Similar findings were also previously reported in another study conducted in England by Turner et al. [56], highlighting the need for targeted interventions to address healthcare inequalities in GCC countries and worldwide in general. The lack of equitable healthcare delivery, especially between rural and urban patients attending PHC centers, calls for measures addressing disparities in both access to, and the quality of, care.

The literature has shown a negative correlation between waiting time and patient satisfaction [57,58]. This is consistent with our findings emphasizing the significant impact of waiting times on patient satisfaction, with long waiting times leading to dissatisfaction. Integrated systems addressing issues related to appointment scheduling, queuing software, and other influencers of waiting time, as reported by Almomani et al. [25], led to a significant reduction in waiting times and improved satisfaction. In China, the integrated health information system also led to a significantly decreased average monthly length of waiting time by 3.49 min (*p* = 0.003) and 8.70 min (*p* = 0.02) for prescription fillings. For prescription fillings, the trend shifted from a slight increase at the start to a significant decrease later on (*p* = 0.003) [57].

The studies in our review focused on the effectiveness of emergency medical services (EMS) as a critical aspect of patient safety and care quality. Patients’ lack of knowledge about the dispatcher number and dissatisfaction with the presence of a male paramedic in cases involving women highlight potential gaps in emergency care services. This underscores the need for targeted efforts to improve awareness and ensure gender-sensitive emergency care. While this systematic review provides valuable insights into patient perceptions of healthcare quality across various dimensions, the lack of data on efficiency represents a literature gap in the comprehensive assessment of healthcare quality in GCC countries. Future studies should explore efficiency-related factors to ensure a holistic understanding of patient experience. However, previous studies have shown that patients who receive tailored, personalized, and holistic care are more likely to be satisfied with delivery efficiency and hospital services, influenced by their willingness to adhere to the agreed-upon treatment plan and course of action [59,60]. Moreover, an efficient admission process was found to be associated with high satisfaction among patients [61,62].

### 4.1. Strengths and Limitations

The issues identified in the current review, including communication problems, impact efforts for the improvement in healthcare quality in GCC countries. Several studies identified effective provider–patient communication as essential to the implementation of value-based healthcare (VBHC) [56,57]. As such, effective patient–provider communication is key to identifying and addressing patient- and family-care needs. With regard to VBHC, the current review found that patients in GCC countries value clear explanations, timely and accurate health information, respectful language, trusting relationships, and being treated with dignity during their encounters with physicians and doctors [25,35]. Therefore, providers in GCC countries should consider these patient needs when implementing patient-centered care and VBHC.

The lack of relevant information on efficiency in these selected studies is concerning, given that access is dictated by healthcare costs. Such a research gap could be associated with challenges in measuring healthcare costs, including the involvement of different payers. Cost is an important factor in VBHC, because this new care-delivery model aims to improve patient outcomes while reducing healthcare costs through efficient care. VBHC lowers care costs by ensuring that all care decisions and activities are consistent with the goals and outcomes of patient care, thus reducing unnecessary procedures and interventions. Efforts to improve efficiency and quality-improvement initiatives in GCC countries should be prioritized. Healthcare delivery efficiency is critical to providing timely and effective care, which contributes to overall patient satisfaction and positive health outcomes.

A significant limitation of this review is that the results may not be representative of all GCC countries because the majority of studies 90% were conducted in Saudi Arabia. Therefore, generalizations based on our review should be made with caution. The results should only be regarded as an indication of potential issues in the healthcare systems of GCC countries from the patient’s perspective. Moreover, several confounding factors, such as education, income, and location, can influence the perceptions and satisfaction of patients with healthcare systems. Moreover, most studies failed to control for such confounding factors, which affected the findings. In addition, we cannot exclude selection bias, despite taking measures to diminish it by having two researchers perform the selection and discuss the differences. Another limitation of this study is that it also limited the search to academic literature in the English and Arabic languages. Therefore, we were unable to review publications in other languages.

### 4.2. Implications for Policy, Practice, and Research

In summary, this review identified several patient-reported quality indicators of importance to individuals in GCC countries, including waiting time, access to health care, lack of patient-care-provider behavior (e.g., language and respect), the built environment, and communication aspects (e.g., access to health information and appointments). Therefore, stakeholders in the healthcare sector, including policymakers and healthcare providers, should focus on these indicators when implementing improvement initiatives for healthcare quality in GCC countries. For instance, policymakers should emphasize high-quality communication with patients, including ensuring alignment with the information needs, culture, language, and education status of patients [11,58]. The disparities in satisfaction across quality domains highlight the significance of personalized, targeted, and context-specific approaches to improving patient-centered care. Addressing safety concerns, promoting equity, reducing wait times, and providing effective emergency care are all critical steps toward improving overall healthcare quality. For future research, the focus should be on the areas identified in the current study as problematic with respect to care quality in the GCC region. Specifically, issues of equity, patient-centered care, and access to healthcare are still at the core of quality concerns in the GCC countries. Thus, future studies can focus on identifying effective strategies for improving care quality by focusing on those particular subjects. Additionally, disparities in quality care and patient satisfaction across the GCC region are an indication of inequalities in care delivery and access to resources across the region. Further research is needed to determine the best strategies for reducing such discrepancies in care.

## 5. Conclusions

The findings of this systematic review contribute valuable insights into patients’ perceptions of healthcare quality in Saudi Arabia, aligning with and expanding upon existing literature evidence. The comprehensive exploration of various dimensions underscores the complexity of patient experiences and highlights key areas for improvement in the delivery of patient-centered, safe, equitable, timely, and effective healthcare. Therefore, healthcare providers and policymakers in GCC countries can use these patient-reported dimensions to plan, assess, and improve care delivery. The identified gaps, particularly in the efficiency domain, provide opportunities for future research and for quality improvement efforts to further enhance the overall quality of healthcare services. Based on the current review, most studies related to the research topic were conducted in Saudi Arabia. Therefore, further studies in other GCC countries can expand the current understanding of this topic.

## Figures and Tables

**Figure 1 healthcare-12-00315-f001:**
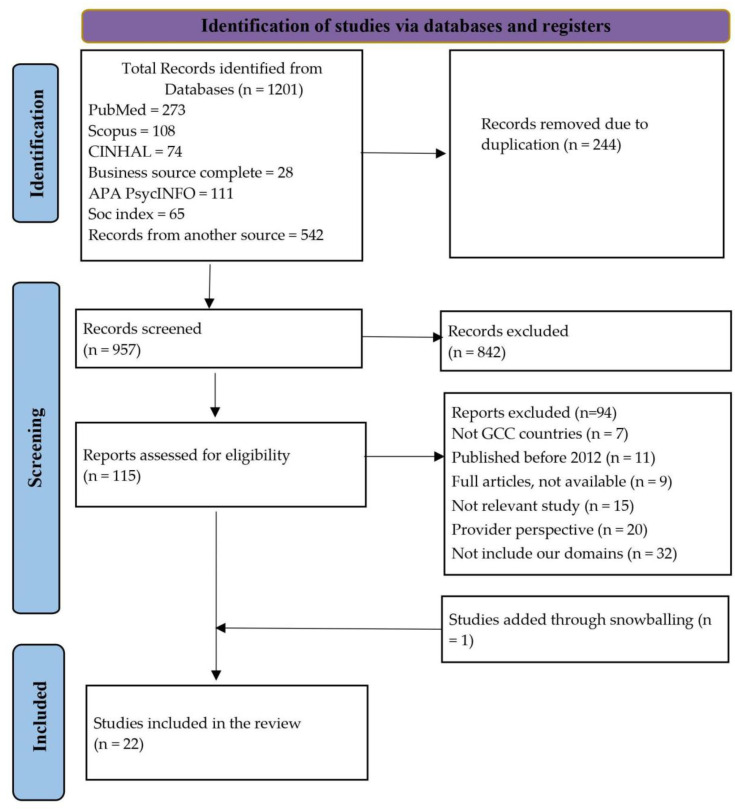
PRISMA flowchart.

**Table 1 healthcare-12-00315-t001:** Search terms.

Domain	Search Terms
Quality	Quality of care, healthcare quality, safe, effective, patient-centered, timely, efficient, and equitable
Patient Centeredness	Person-centered care, patient-centeredness, family-centred care, people-centered care, relationship-centered care, client-centered care, person-directed care, and individualized care
Gulf Cooperation Council Countries	Saudi Arabia, Kuwait, the United Arab Emirates, Qatar, Bahrain, and Oman

**Table 2 healthcare-12-00315-t002:** Inclusion and exclusion criteria.

Criteria	Inclusion criteria	Exclusion criteria
Population	GCC countries	Non-GCC countries
Intervention	Quality of healthcare services from the patient viewpoint	Quality of healthcare services from the healthcare professional viewpoint
Comparison	Correlation between the Gulf Cooperation Council countries and healthcare quality	Not applicable
Outcome	IOM domains: safety, timeliness, efficiency, effectiveness, equitability, and patient-centeredness	Other domains
Time of publication	Articles published from 2012 to 2023 Peer-reviewed scholarly articles English and Arabic articles	Articles published before 2012 Reports Duplicate publications Letters, Editorials Non-peer-reviewed studies Other languages

PICOT framework: patient, intervention, comparison, outcome, and time.

**Table 3 healthcare-12-00315-t003:** Characteristics of the publication reviewed, n = 22.

Classification Category	Subcategory	N (%) *	Reference Index (See Table A4 in Appendix B)
Year of publication	2022 to 2023	4 (18%)	3, 4, 5, 28
	2020 to 2021	4 (18%)	7, 11, 12, 14
	2018 to 2019	2 (9%)	9, 13,
	2016 to 2017	3 (14%)	6,19,26
	2014 to 2015	4 (18%)	21, 22, 24, 27
	2012 to 2013	5 (23%)	1, 2, 8, 15, 20
Research approach	Qualitative	2 (9.1%)	12, 26
	Quantitative	19 (86.4%)	1, 2, 3, 4, 5, 7, 8, 9, 11, 13, 14, 15, 19, 20, 21, 22, 24, 27, 28
	Mixed approach	1 (4.5%)	6
Method of data collection/design	Survey	18(82%)	1, 2, 3, 4, 5, 7, 8, 9, 11, 13, 14, 15, 19, 20, 21, 22, 24, 28
	Interviews	3 (13.5%)	12, 26, 27
	Survey and interview	1 (4.5%)	6
Quality dimensions	Safety	2 (9%)	5, 14
	Timeliness	6 (27%)	6, 9, 13, 24, 27
	Efficiency	0	-
	Effectiveness	1 (4.5%)	26
	Equitability	1 (4.5%)	19
	Patient-centeredness	20 (90%)	1, 2, 3, 4, 5, 6, 7, 8, 9, 11, 12, 14, 15, 19, 20, 21, 22, 24, 26, 28
Country	Saudia Arabia	20 (90%)	1, 2, 3, 4, 5, 6, 7, 8, 11, 12, 13, 14, 15, 19, 20, 21, 22, 24, 27, 28
	Oman	0	-
	Qatar	0	-
	Bahrain	0	-
	United Arab Emirates	2 (10%)	9, 26
	Kuwait	0	-
Setting	Inpatient	4 (18%)	1, 2, 14, 26
	Outpatient	15 (68%)	3, 4, 5, 6, 7, 8, 9, 11, 13, 19, 20, 21, 22, 24, 28
	Online	3 (14%)	12, 15, 27
Sample size	Fewer than 100 respondents	1 (5%)	12
	100 to 500 respondents	12 (54%)	1, 2, 3, 6, 8, 13, 14, 20, 21, 24, 28
	501 to 1000 respondents	8 (36%)	4, 5, 7, 11, 15, 19, 22, 26
	More than 1000 responders	1 (5%)	27
Critical appraisal (JBI) quantitative	Good quality	18 (82%)	1, 2,3, 5, 7, 8, 9, 11, 13, 14, 15, 19, 20, 21, 22, 24, 27, 28
	Fair quality	1 (4.5%)	4
Critical appraisal (CASP) qualitative	Good quality	1 (4.5%)	12
	Fair quality	1 (4.5%)	26
Mixed method appraisal tool (MMAT)	Good quality	1 (4.5%)	6
	Fair quality		

* Percentage of the 22 publications for each subcategory.

**Table 4 healthcare-12-00315-t004:** Systematic map.

Problem Category	Problem Identified	Improvement Measures
Safety	Medication errors Diagnostic errors	Feeling of safety. Sense of neatness and cleanliness.
Timeliness	Scheduling issues, prolonged waiting time, and missing flow problems.	Use of private means for transportation to hospitals.
Effectiveness	Emergency services shortcomings	Response-strengthening Initiatives
Equity	Healthcare disparities	Equity-enhancement Strategies
Patient-centeredness	Patient-care challenges	Patient-care Enhancement

**Table 5 healthcare-12-00315-t005:** Summary of conclusions.

Quality Domain	Conclusions
Safety	Diagnostic errors, communication issues, and medication errors contribute to increased financial expenditure, increased care needs, more physical health issues, and mental health issues.
Timeliness	Notable issues included appointment problems in the scheduling software, early patient arrivals, missing flow problems, and ticket numbering problems.
Efficiency	No reported findings.
Effectiveness	Deficits were reported in emergency medical services.
Equity	Lack of equitable care delivery, especially between the rural and urban PHC patients in terms of dignity, respect, and comprehension of the treatment.
Patient-centered care	Issues identified for patient-centered care included accessibility, reduced cost, information, and staff attitudes. Other aspects included staff competence, physical comfort, and privacy, among others. Most patients expect high satisfaction with healthcare quality.

## Data Availability

Data relevant to the study are either included in the article or uploaded as additional files. On reasonable request, detailed ratings of methodological quality can be provided.

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
