# Peer review of "Healthcare Quality from the Perspective of Patients in Gulf Cooperation Council Countries: A Systematic Literature Review"

_healthcare, 2024, doi:10.3390/healthcare12030315_

Round 1

Reviewer 1 Report

Comments and Suggestions for Authors

Through literature review, this paper attempts to obtain the quality of health care in GCC countries from the perspective of patients. The whole thing is interesting. Some issues require further explanation and minor modifications to improve the quality of the manuscript.

1. It can be seen from the abstract that the authors have identified many problems in healthcare from the perspective of patients in GCC countries through literature review. These problems do not have a detailed classification at present, and seem to be relatively scattered. This is still relatively lacking in actual guidance, and should be condensed and classified.

2. Keywords should highlight the characteristics of the research. Re-think keywords and revise them.

3. Line 98: What are the 6 domains? Although described in line 52, there is no indication that it is being used here. As mentioned in the introduction, quality is a complex concept. It is also worth thinking about how to distinguish these six aspects and their coupling effects in the quality research done in this paper.

4. There are many evaluation methods, why choose PRISMA. What are the outstanding advantages of this method compared with other methods?

5. The results of the study lacked a systematic map to show the problems identified and measures for improvement. It would be nice to add a system diagram if possible.

6. What are the commonalities and individual characteristics between the problems of the GCC countries and those of other countries? Whether the author has thought. Why is the research object of literature review limited to GCC countries? It is suggested to supplement the characteristics of this country in the introduction, and to compare with other existing and up-to-date literature to explain the necessity and significance of these studies.

Reviewer 2 Report

Comments and Suggestions for Authors

The article entitled "..." is within the scope and focus of the journal.

The abstract fulfils the requirements for research of this kind. 

It is an interesting study with relative relevance to the specific area. 

The introduction is well-structured and substantiated with recent authors who are appropriate to the subject. 

The methodology should detail the stages of the review (and what each one contributes to the manuscript and research in general). 

In formal terms, the document should be generally revised and standardised according to the template used by the journal in question. There are some problems with the way the article fits into the template (with spacing that is not standardised). 1. between paragraphs, points and sub-points of the table of contents (e.g. 3.33, 3.3.4, etc.). 2. some authors are still included in the body of the text. They should be removed so that they are like the vast majority of others who are well listed and referenced in the body of the text in relation to the bibliographical references (e.g. 302/3); 3. In table 2 there are words that are split up and hinder the reader's comprehension. Table A1 Results of CASP Qualitative Checklist - Total Points" has text that is unnoticeable and uncharacterised. The documents appended to the manuscript are, provided they are properly presented, very important for understanding the study and the purposes of the research. 

The specific conclusions should also be included in a table or summary table where the main conclusions of the study can be clearly and objectively visualised and understood.

Reviewer 3 Report

Comments and Suggestions for Authors

The research motivation is not strong. Why is there a need to conduct Healthcare Quality in GCC?

Please use some diagrams to show the linkage between the authors, variables, outcomes of the study.

Why Table 1, 2 then jump to Table 7?

A systematic literature review serves as a foundation for future research by summarizing the current state of knowledge on a specific topic. Researchers can build upon the findings of the review to design and conduct new studies. So, what is the future research can be done with this literature?

Comments on the Quality of English Language

Minor revision is required. 

Round 2

Reviewer 2 Report

Comments and Suggestions for Authors

Dear editors and authors, 

The changes made by the authors fully respond to my reflections and suggestions. 

Sincerely